# Neck Circumference as a Valuable Tool to Identify the Risk of Metabolic Syndrome in Mexican Children

**DOI:** 10.3390/children11080908

**Published:** 2024-07-27

**Authors:** Evelyn Valencia-Sosa, Guillermo Julián González-Pérez, Clío Chávez-Palencia, María Guadalupe Vega-López, Enrique Romero-Velarde

**Affiliations:** 1Doctorado en Ciencias de la Salud Pública, Centro Universitario de Ciencias de la Salud (CUCS), Universidad de Guadalajara, 950 Sierra Mojada St., Guadalajara 44340, Jalisco, Mexico; evelyn.valencia@edu.uag.mx (E.V.-S.); guadalupe.vega@academicos.udg.mx (M.G.V.-L.); 2Departamento Académico de Disciplinas Especializantes de Ciencias de la Salud I Área de Nutrición, Universidad Autónoma de Guadalajara, Av Patria 1201 Lomas del Valle, Guadalajara 45129, Jalisco, Mexico; 3División de Ciencias de la Salud, Centro Universitario de Tonalá, Universidad de Guadalajara, Av. Nuevo Periférico 555 Ejido San José Tatepozco, Tonalá 45425, Jalisco, Mexico; clio.chavez@academicos.udg.mx; 4Instituto de Nutrición Humana, Universidad de Guadalajara, Salvador Quevedo y Zubieta 750, Guadalajara 44340, Jalisco, Mexico; enrique.rvelarde@academicos.udg.mx

**Keywords:** obesity, neck circumference, body mass index, metabolic syndrome, waist circumference

## Abstract

Background/Objectives: Neck circumference (NC) has been proposed as a simple measurement to identify patients with overweight and obesity. It has been found that adipose tissue at the cervical level is associated with the presence of metabolic alterations. The aim of this study was to estimate the association between NC and indicators of Metabolic Syndrome (MS) to subsequently estimate its capacity to identify the risk of MS compared to waist circumference (WC) and Body Mass Index (BMI). Methods: A cross-sectional study was carried out with a sample of 286 children 6–9 years old who attended six public primary schools in Jalisco, Mexico. Pearson’s correlation coefficients along with sensitivity and specificity tests were performed to analyze the relationship between NC and MS indicators. Odds ratio (OR) and concordance analyses were performed considering the Kappa index. Results: NC showed statistically significant correlations with all MS indicators except for LDL cholesterol and total cholesterol. The cut-off points of NC to identify MS according to sex was >27.4 cm for girls and >29.8 cm for boys. The association of NC with values above the cutoff point and the presence of MS was OR: 21.6 (CI: 7.11–65.74). Conclusions: NC represents a simple and cost-effective alternative to identify children at risk of MS when compared to BMI and WC.

## 1. Introduction

Metabolic syndrome (MS) consists of a series of metabolic alterations whose simultaneous presence increases the risk of developing type II diabetes mellitus and cardiovascular diseases [1]. Generally, among the indicators considered for its identification are the following: alteration of blood pressure (BP), presence of obesity, and alterations in blood glucose and lipid levels. Nevertheless, the diversity in diagnostic criteria has made it difficult to obtain accurate data regarding its prevalence, especially in the pediatric stage [2]. In this sense, the International Diabetes Federation (IDF) and the National Cholesterol Education Program (NCEP) Adult Treatment Panel III (ATP-III) are the most used criteria for identifying MS in children [3]. However, IDF does not have an age-specific reference values for children under 10 years old.

Despite the absence of a consensus on indicators and cutoff points to include, most authors agree that excess body fat plays a significant role in the development of MS [4]. Among the classic anthropometric indices and measurements used to identify individuals with obesity or adiposity are body mass index (BMI), considered in the WHO criteria, and waist circumference (WC), which is considered more useful by other authors [5]. It is noteworthy that these indicators hold a series of disadvantages. On the one hand, BMI is the most commonly used parameter for the diagnosis of overweight and obesity, but it does not differentiate between the presence of fat or its distribution. On the other hand, WC indicates the presence of adiposity at the abdominal level, which represents a greater health risk; however, there is no standard measurement method, and various practical obstacles may arise when measuring it properly.

In this sense, NC has been proposed as a simple measurement to identify overweight and obesity. Furthermore, it has been found that adipose tissue at the cervical level is associated with the presence of metabolic alterations because it is closely related to visceral fat [6]. It has been reported that NC is easier to perform than WC and simpler to interpret than BMI, so it is important to verify its utility within the criteria to identify MS and, thus, facilitate the detection of individuals at risk. Therefore, the objective of this study is to estimate the association between NC and indicators of MS to subsequently estimate its capacity to identify the risk of MS compared to WC and BMI.

## 2. Materials and Methods

### 2.1. Participants

The sample in this study is derived from a larger research project entitled “Participatory Intervention to Improve Nutrition and Physical Activity in Acatlán de Juárez, Jalisco”, which was conducted in six public primary schools in Acatlán de Juárez and Villa Corona, Jalisco, Mexico [7]. Inclusion criteria consisted of boys and girls attending 1st-4th grade in elementary schools with informed consent signed by their parents or tutors. Exclusion criteria were children suffering or having a history of any cardiovascular and metabolic diseases, asthma, physical disabilities, or active consumption of medications. 

For sample size calculation, the criteria of a school intervention study conducted by Li et al. in 2014 [8] were used, considering a type I error of 0.05, power of 80%, and 6 clusters. A 30% loss to follow-up was assumed, resulting in a total sample size of 288 children from six schools. One group was randomly selected from each grade level of these schools, and within each group, 12 children were randomly chosen (48 children in total per school).

### 2.2. Anthropometric Measurements

Anthropometric measurements were taken in duplicate and carried out with prior training of the evaluators using the Habitch method. 

NC was measured at the midpoint level of the thyroid cartilage and perpendicular to the longitudinal axis of the neck. The subject remained upright, facing forward with the head in the Frankfort plane, and breathing normally [9]. WC measurement was performed with the subject standing upright, hands crossed over the chest, and the area uncovered. It was registered at the end of exhalation [10]. For the circumference measurements, a Lufkin model W606PM(Apex Tool Group, TX, United States) metal tape with an accuracy of 1 mm was used.

A detailed explanation of the remaining anthropometric measurements was published in a previous article corresponding to the aforementioned project [7].

### 2.3. Biochemical and Clinical Measurements

For blood sample collection, parents and guardians were reminded to ensure that children fasted from calorie-containing foods and beverages for 12 h prior to the sampling. Parents and guardians were also informed about the general specifications to consider for the anthropometric and biochemical assessment. Measurements were taken during the early hours of the school schedule (between 8:00 and 10:00 a.m.), and approximately 10 mL of blood per child was withdrawn.

The determination of blood chemistry analytes was carried out by spectrophotometry in liquid chemistry. Hematological biometry was performed using the SISMEX equipment (Sysmex, Kobe, Japan) by impedance. The laboratory where the assays were performed is enrolled in the external Quality Control PACAL (Quality Assurance Program), a company certified in ISO 9001. Additionally, it holds a Laboratory Operation Notice before the Health Secretary of Jalisco and the University of Guadalajara. They are registered with SEMARNAT (Secretary of Environment and Natural Resources) and have transportation and reception of hazardous biological waste with the company Corporativo Techno Ambienta, Lerma de Villada, Mexico.

On the other hand, BP was taken by the evaluators in accordance with the American Heart Association procedure for children.

### 2.4. Variable Classification

The criteria proposed by the Expert Panel of the National Cholesterol Education Program of the United States (ATPIII) for the pediatric population were used, including the presence of 3 or more of the following indicators: abdominal obesity with WC above the 90th percentile, BP above the 90th percentile, triglycerides >110 mg/dL or above the 95th percentile, HDL cholesterol <40 mg/dL or below the 5th percentile [11].

The participant’s weight classification was determined using the BMI for age z-score with the following cutoff points: underweight < −2 SD, normal weight between −2 and +1 SD, overweight between +1 and +2 SD, and obesity > +2 SD [12].

### 2.5. Statistical Analysis

The Kolmogorov–Smirnov test was performed to determine the normality of all variables. Quantitative variables were expressed as mean and standard deviation, whereas qualitative variables were expressed as frequency and percentage. Student’s t-test was conducted for comparisons between groups with quantitative variables, and the χ^2^ test was used for comparisons of qualitative variables. Pearson correlation coefficients were also calculated between NC, WC, BMI, and biochemical and clinical variables.

To calculate cutoff points for MS based on WC and NC, sensitivity and specificity tests were performed. Additionally, the association between NC, WC, BMI, and MS (based on the classification and according to cutoff points) was analyzed. Odds ratios were calculated, with a 95% confidence interval. Various statistical programs were used for data analysis: EPIDAT version 4.2 was used to calculate the Kappa coefficient and difference in proportions [13] and IBM SPSS version 22.0 was used for descriptive analysis, correlation calculation, and linear regressions. MedCalc version 9.4.2.0 was employed for sensitivity and specificity tests.

## 3. Results

A total of 286 schoolchildren aged 6 to 9 years were included, with an average age of 7.5 years. Of the total sample, 47.9% (*n* = 137) were girls and 52.1% (*n* = 149) were boys. 

Table 1 shows the mean and standard deviation of anthropometric and biochemical variables by sex. In most variables, boys displayed higher values compared to girls, except for % body fat, triglycerides, total cholesterol, and LDL cholesterol. However, only differences in glucose and NC were statistically significant.

The prevalence of overweight and obesity was 15.0% and 24.8%, respectively. As for MS in the entire cohort, it was observed in 8.6% of children. Girls showed a higher prevalence (10.9%) compared to boys (6.7%). Table 2 shows the prevalence of impaired biochemical parameters by age. It can be observed that more than half of the children had HDL cholesterol levels below the recommended values. BP levels also displayed a higher frequency of altered values compared to other indicators, especially SBP. 

Correlations between NC and MS indicators were statistically significant except for total cholesterol (*p* = 0.386) and LDL cholesterol (*p* = 0.458). WC displayed a similar pattern, as both total cholesterol (*p* = 0.139) and LDL cholesterol (*p* = 0.09) showed non-significant values. On the other hand, BMI showed high correlations with triglyceride (*p* < 0.001) concentrations and SBP levels (*p* < 0.001), and just as NC and WC, it did not show significant values with total cholesterol and LDL cholesterol levels. Overall, the three anthropometric variables showed a similar pattern in terms of the magnitude and significance of correlations (Table 3).

Regarding sensitivity and specificity tests, the global cutoff points to identify MS based on NC were >27.4 cm for girls and >29.8 cm for boys, with an area under the curve greater than 0.8 in both cases. Regarding WC, the general cutoff points to identify MS were >62.5 cm for girls and >69.5 cm for boys, with the area under the curve values greater than 0.8 in both cases. Table 4 shows the cutoff points for both measurements by sex and age.

According to the association analyses, children with an NC above the specified cutoff point had a 21 times higher risk of presenting MS than those with an NC below the cutoff point. Regarding WC, the risk of presenting MS was 28 times higher in children with a value above the specified cutoff point. As for BMI, where children were classified as overweight or normal weight, the risk of presenting MS was 12 times higher (Table 5).

## 4. Discussion

The main objective of the present study was to demonstrate the relationship between NC and MS components in the pediatric stage and estimate its capacity to identify MS risk compared to WC and BMI. The anthropometric variables were similar in girls and boys except for NC, where a higher value was observed in boys with statistically significant differences; this pattern has been reported by several authors [14,15,16]. Coutinho et al. [17] attribute this outcome to a greater muscle mass in this area, compared to females. Regarding biochemical variables, girls showed higher values in most parameters; however, the differences were not statistically significant and do not appear to hold clinical relevance. A prevalence of overweight and obesity of 39.8% was found. This figure is higher than the national average reported (37.4%) according to the 2021 Health and Nutrition Survey in Mexico [18]. 

Regarding biochemical variables, it was found that HDL cholesterol was the indicator that showed the highest alteration, as more than half of the participants had values below the recommended range. In contrast, blood glucose levels showed normal values in the entire sample. Kurtoglu et al. [14] reported a similar trend finding low HDL cholesterol values compared to other indicators. On the other hand, our data differ from those presented by González-Cortés et al. [19] as in their study, BP followed by triglyceride concentrations were the indicators that showed the most significant alterations.

The prevalence of MS was 8.6% in the studied sample, with a higher prevalence in girls compared to boys (10.6% vs. 6.7%). This fact may be related to the higher prevalence of overweight and obesity found in girls. These values differ from the data presented in similar research. On the one hand, Hassan et al. [20] found a prevalence of 52% using the ATP III criteria; however, their sample consisted only of obese patients. On the other hand, Kurtoglu et al., in 2012 [14], found a prevalence of MS of 24.1% using the International Diabetes Federation (IDF) criteria in a sample composed of overweight and obese children. It is worth mentioning that few studies report the prevalence of MS in their sample as they focus on analyzing associations by indicator, and most include only participants with excess weight.

The correlation analysis between NC and MS indicators showed statistically significant associations with glucose concentrations (r = 0.21, *p* < 0.001), triglycerides (r = 0.25, *p* < 0.001), HDL cholesterol (r = −0.18, *p* < 0.003), SBP (r = 0.36, *p* < 0.001), and DBP (r = 0.23, *p* < 0.001). Of these, DBP was the variable that showed the highest values. These findings are consistent with those reported by Gonzáles-Cortés et al. [19], where correlations between NC and triglyceride concentrations were found to be r = 0.25 (*p* < 0.0001) in boys and r = 0.31 (*p* < 0.0001) in girls. Regarding SBP, the values were r = 0.27 (*p* < 0.0001) for boys and r = 0.33 (*p* < 0.0001) for girls. Similarly, Peña-Valdez et al. [21] reported correlations of r = 0.20 for triglycerides (*p* < 0.057) and r = 0.54 (*p* < 0.001) for SBP. Castro-Piñero et al. [22] and Kelishadi et al. [23] showed a similar trend in terms of the reported correlations with triglycerides and SBP. The correlations found in this study align with what has been reported in most articles relating NC to biochemical and clinical variables, as generally, the r values tend to be higher and statistically significant when it comes to BP levels, while correlations showed contradictory and non-significant values when analyzing total cholesterol and LDL cholesterol concentrations [24].

It is important to mention that although the correlations found in this study between NC and MS indicators showed r values below 0.5, these values are similar to those reported with WC. This similarity also occurs when analyzing the risk associated with having a higher neck circumference or WC (OR: 21, *p* < 0.001 vs OR: 28, *p* < 0.001) for presenting MS; although the risk is higher with elevated WC, the value does not differ significantly from what is reported with NC, unlike the risk reported with elevated BMI (OR: 12, *p* < 0.001). According to the results found in this study, NC could be a useful tool to identify children with risk for MS, as the area under the curve values in sensitivity and specificity tests were greater than 0.8 regardless of sex and age group. According to the literature, an area under the curve value between 0.75 and 0.85 has moderate accuracy, while a value above 0.85 has high accuracy. Taking these data into account, it is confirmed that NC has an accuracy considered high for the identification of MS [25], and, taking these data into account, it is confirmed that NC has a precision considered high for the identification of MS.

In the case of girls, an NC above 27.4 cm would indicate the presence of MS, whereas for boys, the value would need to be above 29.8 cm. The values reported by Gomez-Arbelaez et al. [15] show a similar cutoff point for boys (29 cm) but a higher value for girls (28.5 cm). It is worth mentioning that the sample in their study consisted of Colombian school-aged children and they were included regardless of their BMI classification. However, they used a modified version of the National Health and Nutrition Examination Survey (NHANES) to diagnose MS. 

In a cohort of Brazilian children, Goncalvez et al. [26] reported that a cutoff above 28.8 cm in girls and 30.4 cm in boys would indicate cardiovascular risk. However, in their study, they did not use specific criteria to evaluate cardiovascular risk as a whole, and they only included participants aged 10 to 14 years. On the other hand, Kurtoglu et al. [14] reported cutoff points of 36 cm for boys and 35 cm for girls; however, their sample only consisted of subjects with obesity. It is important to note that since most studies include only participants with excess weight, the reported cutoff points are higher than those found in this study.

Another point worth highlighting is the comparison between WC, BMI, and NC and their relationship with MS indicators. In some statistical tests, WC showed higher values compared to NC; however, the values did not differ significantly, in contrast with BMI where NC showed better results. This aspect is important because currently both WC and BMI are considered indicators to identify MS according to various classifications. In this case, our results suggest that NC has a similar or even greater association, in the case of BMI, with MS indicators. Therefore, it could be considered for inclusion in the criteria in further studies. While it cannot be said that using only NC can diagnose MS, there would be a high probability of identifying individuals at risk, especially when infrastructure does not allow for more specific tests.

The advantages of using NC over other anthropometric measurements are not limited to practical aspects. It is not only easier to perform, more cost-effective, and less invasive, but it also considers fat distribution in contrast to BMI. It has been previously reported that fat located in the upper body poses a greater health risk due to the release of free fatty acids [27]; additionally, cervical fat is responsible for releasing >50% of these molecules, making it the major contributor of free fatty acids to circulation, which are implicated in most metabolic alterations [28,29]. Arias-Tellez, et al. [30] demonstrated that adipose tissue at the cervical level is associated with a higher cardiovascular risk and a proinflammatory state in young adults. The associations they reported were independent of BMI, fat percentage, and visceral adipose tissue, suggesting that adipose tissue at the cervical level is as valuable as visceral adipose tissue in predicting cardiovascular risk.

It is important to mention that the present study has some limitations since the studied sample was obtained from a large-scale project, so the methodological part could not be controlled at the time of data collection. Another limitation derived from this last point is that the sample size might not be sufficiently large to draw general conclusions. However, multiple authors have approached NC in a similar manner, with a sample size of 300 children or less delivering interesting outcomes [20,21,26,31,32,33,34].

Additionally, the proposed cutoff points are limited to the studied population. Nonetheless, the results found are consistently valuable to consider further exploration of NC as an option to identify children at risk of MS with even larger samples.

## 5. Conclusions

NC represents a simple and cost-effective alternative to identify children at risk of MS when compared to BMI and WC. Although more research is needed to suggest its inclusion in the diagnosis criteria, the data provided herein indicate that it could be a valuable tool for clinical assessment.

## Figures and Tables

**Table 1 children-11-00908-t001:** Mean and standard deviation of anthropometric, biochemical, and clinical variables.

Variable	Girls (*n* = 137)		Boys (*n* = 149)		*p*
	Mean	SD	Mean	SD	
Age (years)	7.5	1.1	7.5	1.2	0.73
Weight (kg)	28.9	9.6	29.6	9.8	0.48
Height (cm)	127.0	9.3	127.2	8.8	0.87
Waist circumference (cm)	59.3	9.4	60.4	9.6	0.33
Neck circumference (cm)	26.6	2.5	27.8	2.6	<0.001
BMI (kg/m^2^)	17.6	3.7	17.9	4.1	0.37
BMI (z)	0.6	1.4	0.8	1.7	0.41
Glucose (mg/dL)	76.5	7.5	79.4	7.9	0.001
Triglycerides (mg/dL)	78.5	36.0	73.7	47.9	0.34
CHOL (mg/dL)	143.7	20.7	139.1	23.4	0.08
HDL-C (mg/dL)	46.6	10.5	46.9	10.1	0.76
LDL-C (mg/dL)	81.5	18.1	77.5	19.3	0.07
SBP (mmHg)	100.0	11.3	101.5	11.8	0.27
DBP (mmHg)	61.8	9.4	62.2	11.6	0.72

BMI: body mass index, CHOL: total cholesterol, HDL-C: high-density cholesterol, LDL-C: low-density cholesterol, SBP: systolic blood pressure, DBP: diastolic blood pressure.

**Table 2 children-11-00908-t002:** Prevalence of impaired biochemical parameters and BMI by age.

Age	High BMI *n* (%)	High FBG *n* (%)	High TG *n* (%)	High CHOL *n* (%)	Low HDL-C *n* (%)	High LDL-C *n* (%)	High SBP *n* (%)	High DBP *n* (%)
6	25 (32.0)	0 (0.0)	10 (12.8)	4 (5.1)	46 (59.0)	2 (2.6)	17 (21.8)	13 (16.7)
7	23 (36.5)	0 (0.0)	9 (14.3)	0 (0.0)	37 (58.7)	0 (0.0)	20 (31.7)	15 (23.8)
8	27 (42.9)	0 (0.0)	8 (12.7)	1 (1.6)	35 (55.6)	1 (1.6)	18 (28.6)	17 (27.0)
9	39 (47.6)	0 (0.0)	13 (15.9)	5 (6.1)	46 (56.1)	0 (0.0)	19 (23.2)	12 (14.6)

BMI: body mass index, FBG: fasting blood glucose, TG: triglycerides, CHOL: total cholesterol, HDL-C: high-density cholesterol, LDL-C: low-density cholesterol, SBP: systolic blood pressure, DBP: diastolic blood pressure, SD: standard deviation. High BMI (overweight + obesity): >1SD, High FBG: >100 mg/dL, High TG: >110 mg/dL, High CHOL: >200 mg/dL, Low HDL-C: <40 mg/dL, High LDL-C: >130 mg/dL, High SBP: >percentile 90, High DBP >percentile 90.

**Table 3 children-11-00908-t003:** Correlation coefficients between WC, NC, BMI, and biochemical and clinical variables.

Variable	NC		WC		BMI	
	r	*p*	r	*p*	r	*p*
Glucose	0.21	<0.001	0.14	0.015	0.15	0.010
CHOL	0.05	0.386	0.08	0.139	0.10	0.97
Triglycerides	0.25	<0.001	0.34	<0.001	0.32	<0.001
HDL-C	−0.18	0.003	−0.28	<0.001	−0.27	<0.001
LDL-C	0.04	0.458	0.10	0.090	0.12	0.050
SBP	0.36	<0.001	0.34	<0.001	0.32	<0.001
DBP	0.23	<0.001	0.22	<0.001	0.21	<0.001
WC	0.91	<0.001	----	----	0.95	<0.001
NC	----	----	0.91	<0.001	0.89	<0.001
BMI	0.89	<0.001	0.95	<0.001	----	----

BMI: body mass index, CHOL: total cholesterol HDL-C: high-density cholesterol, LDL-C: low-density cholesterol, SBP: systolic blood pressure, DBP: diastolic blood pressure.

**Table 4 children-11-00908-t004:** Cut-off points of neck and WC to identify MS by gender and age group.

**Neck Circumference**							
**Sex**	**Age**	**n**	**AUC**	**95% CI**	**Cut-Off Point**	**Sensitivity**	**Specificity**
Girls	6–7	70	0.821	0.711 to 0.902	>27.3	85.71	85.71
	8–9	67	0.956	0.875 to 0.990	>29.2	87.50	91.53
Boys	6–7	71	0.826	0.718 to 0.906	>26.7	100.00	62.69
	8–9	78	0.817	0.713 to 0.896	>29.6	83.33	79.17
**Waist Circumference**							
**Sex**	**Age**	**n**	**AUC**	**95% CI**	**Cut-Off Point**	**Sensitivity**	**Specificity**
Girls	6–7	70	0.803	0.690 to 0.888	>58.4	85.71	82.54
	8–9	67	0.942	0.856 to 0.984	>65.1	100.00	76.27
Boys	6–7	71	0.948	0.867 to 0.986	>63.6	100.00	88.06
	8–9	78	0.802	0.696 to 0.884	>69.5	83.33	84.72

AUC: area under the curve, CI: confidence interval.

**Table 5 children-11-00908-t005:** Association between NC, WC, BMI, and MS.

		Metabolic Syndrome		*p*	OR	CI
		Present	Absent			
NC	Present *	21	51	<0.001	21.62	7.11–65.74
	Absent	4	210			
WC	Present *	21	41	<0.001	28.17	9.19–86.34
	Absent	4	220			
BMI	Overweight and obesity	22	92	<0.001	12.75	3.72–43.78
	Normal weight	3	160			

NC: neck circumference, WC: waist circumference, BMI: body mass index, OR: odds ratio, CI: confidence interval, * Presence of MS according to the calculated cut-off points for gender and age group.

## Data Availability

Data supporting the reported results are available upon request to the corresponding author due to privacy and ethical reasons.

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
