# Peer review of "Neck Circumference as a Valuable Tool to Identify the Risk of Metabolic Syndrome in Mexican Children"

_children, 2024, doi:10.3390/children11080908_

Round 1
Reviewer 1 Report
Comments and Suggestions for Authors
Valencia-Sosa et al. estimate the association between NC and indicators of MS and estimate its capacity to identify the risk of MS compared to WC and BMI. This is an interesting topic considering the increased prevalence of metabolic syndrome in children and adolescents, and its long-term cardiometabolic complications. However, there are some questions that must be addressed:
Introduction:
- The authors mention WC and BMI, but there are more anthropometric indicators in the scientific literature related to metabolic syndrome. The authors should better justify why they focus on these two indicators for comparison with neck circumference. In short, a better contextualization of the objective of the study is lacking.
Methods:
- What were the eligibility criteria for participants?
Results are clearly written. Discussion section is well written, but results on differences observed in the baseline characteristics between boys and girls must be discussed.
Conclusions must be improved according to the results reported in the manuscript, and should have a more critical, reflective point of view and propose new future lines of action.
Comments on the Quality of English LanguageNo comments
Author Response
Dear reviewer 1.
Thank you for your highly valuable comments on our manuscript. We reply to every comment here below:
Comment 1: Introduction: the authors mention WC and BMI, but there are more anthropometric indicators in the scientific literature related to metabolic syndrome. The authors should better justify why they focus on these two indicators for comparison with neck circumference. In short, a better contextualization of the objective of the study is lacking.
Response 1: We have moderately modified the introduction text so that it provides a better context on why body mass index and waist circumference are relevant and why they are not as practical as others like neck circumference. However, we would like to highlight that as for the pediatric population, BMI and WC are the two anthropometric indicators that are included in the criteria for diagnosing metabolic syndrome. We have also added a couple of reference to support this rationale (lines 46-51).
Comment 2: Methods: what were the eligibility criteria for participants?
Response 2: We are surprised that we did not notice the eligibility criteria were missing. Thank you for pointing it out. They have been added in the methodology section (lines 70-73).
Comment 3: Results are clearly written. Discussion section is well written, but results on differences observed in the baseline characteristics between boys and girls must be discussed.
Response 3: The baseline characteristics between boys and girls have been added in the first paragraph of the discussion section (lines 194-199).
Comment 4: Conclusions must be improved according to the results reported in the manuscript, and should have a more critical, reflective point of view and propose new future lines of action.
Response 4: Thank you for this suggestion. We believe it was highly relevant. The conclusion has been rewritten to deliver a more critical and reflective point of view (lines 294-297).

Reviewer 2 Report
Comments and Suggestions for Authors
Dear Authors,
Thank you for your interesting study on neck circumference (NC) and metabolic syndrome (MS) indicators in children. Your findings suggest that NC could be a useful tool for identifying MS risk.
I would like to inquire about the sample size used in your study. While 286 children provide helpful insights, is this number enough to make general conclusions and suggest cut-off points for children as your title indicates? Has an expert in statistics contributed to the presentation and interpretation of the data to ensure the results are well-supported?
Additionally, I would suggest improving the clarity of the manuscript and your data presentation to enhance its scientific soundness.
Please consider the specific comments below.
Specific Comments:
Line 2: Please reconsider your title to ensure it reflects whether your research supports the generalization of your results.
Line 41: You state that there is no consensus on cutoff points or indicators to include. Please expand your literature review justifying the variable classification in lines 102-107. Please be more explanatory throughout the text about the metabolic syndrome definition application in your group of children.
Line 50: Add a full stop at the end of the sentence.
Line 64: Add the citation for the study (80).
Line 68: Are the control and intervention groups included in this study or in the larger research project? Please revise. Is the power calculation for this study or for the larger research project?
Table 1: Non-normally distributed variables are usually presented as median and 25th, 75th percentiles.
Lines 136-142: Please rewrite the paragraph and reconsider the following terms:
· Line 136: Replace the term "nutritional status."
· Line 137: Define or replace “general population.”
· Line 138: For individuals aged 12 years and younger, the appropriate terms are "girls" and "boys."
· Line 139: Replace the phrase “altered laboratory values.”
Line 143: Add p-values in the text.
Table 2: Consider using the appropriate term (overweight) as defined by WHO instead of “high BMI.” Consider the same for the rest of the variables. Please include n (number of children) for each age group and variable (usually values are presented as n(%)).
Line 160: Do you mean 29.6 cm? Please clarify.
Table 4: Usually, 95% CI are separated by comma (,), “to,” or hyphen (-).
Line 174: Please provide in the result section a detailed description of the presence of metabolic syndrome in the group of 286 children.
Comments on the Quality of English LanguageWhile I am not a native English speaker, I would suggest that the English in this article needs major revision to be clearer and more scientifically sound.
Author Response
Dear reviewer 2.
Thank you for your highly valuable comments on our manuscript. We reply to every comment here below:
Comments 1: I would like to inquire about the sample size used in your study. While 286 children provide helpful insights, is this number enough to make general conclusions and suggest cut-off points for children as your title indicates? Has an expert in statistics contributed to the presentation and interpretation of the data to ensure the results are well-supported?
Response1: We understand that this is relevant information that should be discussed. It has now been added as a limitation in the discussion section. We confirm that the sample size calculation for this work was not reviewed by an expert in statistics, only for the intervention project this cohort belongs to (cited in the methodology section). In the lines we have added, we mention a few other authors that have approached the same (or at least similar) objectives with a sample size similar to ours in a constructive comparative manner (lines 286-289).
Comment 2: Additionally, I would suggest improving the clarity of the manuscript and your data presentation to enhance its scientific soundness.
Response 2: We appreciate this suggestion. A second expert in the English language has revised the manuscript to improve the scientific soundness.
Comment 3: Line 2: Please reconsider your title to ensure it reflects whether your research supports the generalization of your results.
Response 3: The title has been modified and we believe it now better reflects the generalization of our outcome. We appreciate this suggestion.
Comment 4: Line 41: You state that there is no consensus on cutoff points or indicators to include. Please expand your literature review justifying the variable classification in lines 102-107. Please be more explanatory throughout the text about the metabolic syndrome definition application in your group of children.
Response 4: We have moderately modified the introduction text so that it provides a better context of the diagnosis criteria of metabolic syndrome. Also, we have added basic information about different criteria proposals (lines 42-51).
Comment 5: Line 50: Add a full stop at the end of the sentence.
Response 5: It has been added now.
Comment 6: Line 64: Add the citation for the study (80).
Response 6: The citation has been added (line 70).
Comment 7: Line 68: Are the control and intervention groups included in this study or in the larger research project? Please revise. Is the power calculation for this study or for the larger research project?
Response 7: Thank you for highlighting this ambiguity. We have rewritten this section, so it does not confuse readers. The sample size calculation was performed for the larger project only. For the purpose of this work, we have taken the same sample. We can complement the justification for this comment with the reply to the first comment (lines 74-79).
Comment 8: Table 1: Non-normally distributed variables are usually presented as median and 25th, 75th percentiles.
Response 8: All variables are expressed in mean ± SD because all of them were p>0.05 for the Kologorov-Smirnov equation (lines 119-120).
Comment 9: Lines 136-142: Please rewrite the paragraph and reconsider the following terms:
Response 9: We appreciate the suggestion to rewrite the paragraph in question. Now they have become lines (149-154). Including the terms that you have pointed out for the next comments (10-14)
Comment 10: Line 136: Replace the term "nutritional status."
Response 10: Replaced due to ambiguity.
Comment 11: Line 137: Define or replace “general population.”
Response 11: Replaced due to ambiguity.
Comment 12: Line 138: For individuals aged 12 years and younger, the appropriate terms are "girls" and "boys."
Response 12: Replaced. Thank you
Comment 13: Line 139: Replace the phrase “altered laboratory values.”
Response 13: Corrected, thank you.
Comment 14: Line 143: Add p-values in the text.
Response 14: They have now been added (lines 156-159).
Comment 15: Table 2: Consider using the appropriate term (overweight) as defined by WHO instead of “high BMI.” Consider the same for the rest of the variables. Please include n (number of children) for each age group and variable (usually values are presented as n (%).
Response 15: We appreciate this observation. We have changed the style to “n (%)” as suggested. We believe it is clearer now. Also, we have added a brief explanation in the table legend to clarify what we mean with “high BMI”, in which case it refers to overweight + obesity (lines 163-166)
Comments 16: Line 160: Do you mean 29.6 cm? Please clarify.
Response 16: We have double checked it. In this case, the value in that line represents the cut off point for the general cohort and it correct, whereas in the table they are showed by sex and age. That is why they are not the same number (lines 173-174).
Comments 17: Table 4: Usually, 95% CI are separated by comma (,), “to,” or hyphen (-).
Response 17: We did not notice this error; it has been corrected.
Comments 18: Line 174: Please provide in the result section a detailed description of the presence of metabolic syndrome in the group of 286 children.
Response 18: We have double checked it. In this case, this information is shown in lines (lines 149-151).

Round 2
Reviewer 1 Report
Comments and Suggestions for Authors
All my comments have been addressed correctly. No more comments.
Comments on the Quality of English LanguageNo comments.
Reviewer 2 Report
Comments and Suggestions for Authors
Dear Authors,
Thank you for thoroughly addressing all the provided comments. I appreciate your diligent efforts in improving the manuscript.